# Natural Language Processing Techniques for Text Classification of Biomedical Documents: A Systematic Review

Cyrille YetuYetu Kesiku *, Andrea Chaves-Villota and Begonya Garcia-Zapirain

eVida Research Group, University of Deusto, Avda/Universidades 24, 48007 Bilbao, Spain
* Correspondence: cyrille.kesiku@opendeusto.es

**Abstract:** The classification of biomedical literature is engaged in a number of critical issues that physicians are expected to answer. In many cases, these issues are extremely difficult. This can be conducted for jobs such as diagnosis and treatment, as well as efficient representations of ideas such as medications, procedure codes, and patient visits, as well as in the quick search of a document or disease classification. Pathologies are being sought from clinical notes, among other sources. The goal of this systematic review is to analyze the literature on various problems of classification of medical texts of patients based on criteria such as: the quality of the evaluation metrics used, the different methods of machine learning applied, the different data sets, to highlight the best methods in this type of problem, and to identify the different challenges associated. The study covers the period from 1 January 2016 to 10 July 2022. We used multiple databases and archives of research articles, including Web Of Science, Scopus, MDPI, arXiv, IEEE, and ACM, to find 894 articles dealing with the subject of text classification, which we were able to filter using inclusion and exclusion criteria. Following a thorough review, we selected 33 articles dealing with biological text categorization issues. Following our investigation, we discovered two major issues linked to the methodology and data used for biomedical text classification. First, there is the data-centric challenge, followed by the data quality challenge.

**Keywords:** text classification; biomedical document; natural language processing; biomedical text classification challenges





## 1. Introduction

The focus on text data is increasing day by day in different fields. Generally in the healthcare field, patient information consists mostly of medical texts or notes taken by doctors and nurses. The classification of medical text in the process of extracting knowledge from medical data has gained momentum in recent times thanks to Natural Language Processing techniques. In this technique, the main approach is the recognition of a necessary pattern that explains a fact from the links between words and sentences in a text. These links give a semantic meaning and allow a good understanding of the information in the text. In health, this helps in the rapid search for the causes of a disease and correlates all the causes extracted from the text to predict the disease. Many other problems are treated by following this approach.

Since 2013 until today, NLP research has demonstrated its inescapable capabilities with very relevant models emerging every year probably. Techniques based on neural network architectures, very intuitive in classification and other important natural language processing tasks [1,2]. Many other problems in health care use text classification such as in the International Classification of Diseases (ICD), which is a medical classification list published by the World Health Organization, which defines the universe of diseases, disorders, injuries and other related health conditions as well as the standard of diagnosis classification [3,4].

In this systematic review, we examine the different articles on patient medical text classification from 1 January 2016 to 10 July 2022, in order to identify the relevant challenges in biomedical text classification. The knowledge gained in this study will clearly map out the methodologies and techniques for future research work. In this study, we seek to answer the questions in the Table 1.

**Table 1.** Research questions and purpose.

| | Question | Purpose |
|---|---|---|
| Q1 | What are the best NLP methods used in medical text classification? | To Describe the best methods used in the medical classification framework based on the evaluation metrics. And identify the challenges |
| Q2 | How are medical text classification datasets constructed? | To study the composition and description of medical texts in the classification task. |
| Q3 | In terms of data, what are the most common problems that medical text classification can solve? | To understand and highlight the common problems and challenges addressed in medical text-based problem solving. |
| Q4 | What are the mostly used evaluation metrics of medical document classification? | To identify the different mostly metrics used in medical document classification |

## 2. Material and Methods

The major purpose of our systematic study is to highlight current issues that text classification systems have to cope with in order to analyze biological text information. The insights discovered in this study will be utilized as a starting point for future research in this area. Table 1 outlines the main questions we hoped to address by the conclusion of this research. To achieve this review system, we have merged the methodologies employed by Sofia Zahia et al. in [5] and those by Urdaneta-Ponte et al. in [6]. On the basis of these strategies, we shall produce our review article.

### 2.1. Data Collection

The articles in the databases were chosen using a variety of methodologies and eligibility criteria which are briefly presented in the following subsections. We initially applied the filter of papers collected from various databases, followed by the filters based on the qualifying criteria. Each metric was used to pick publications that were relevant to our research.

#### 2.1.1. Searched Databases

Several databases were used to conduct the literature search, including Web of Science, Arxiv, IEEE, ACM, Scopus, and MDPI. The selection time of articles was limited from 1 January 2016 to 10 July 2022. Several factors influenced our choice of publications, including the search terms, which covered studies published on biomedical text classification as well as image-text classification.

#### 2.1.2. Search Terms

Several terms were used to look for works on biomedical text classification task; some of these terms were combined to enhance the search in multiple databases. The terms chosen for the selections were: "text classification", "medical text", "medical document", "healthcare", "patient records", "text prediction", "nursing notes", "Natural Language Processing", "text-image", "biomedical text classification", "nursing text prediction", "prediction", "classification", "image", "text", "Machine learning", "transformers", "LSTM", "GRU", "clinical" and "clinical text".

### 2.1.3. Inclusion Criteria

The initial stage in the selection procedure was to look through titles and abstracts to discover papers that fulfilled the needed criteria. Then duplicates were removed. Because medical record classification encompasses numerous applications, such as the detection and classification of text in nursing picture notes, relevant matching publications were obtained and classified.

### 2.1.4. Exclusion Criteria

The following exclusion criteria were applied to select the papers: date of publication, type of publication, ranking of the journal in case the paper was published in an international journal, type of problem studied in the paper, and finally the number of citations of the paper.

Figure 1 depicts the revised flowchart for PRISMA in [6]. This systematic review's data gathering approach followed a logical progression until only 33 publications were deemed appropriate for analysis. Each database indicated in Section 2.1.1 as a source of publications was recognized, along with a total of 894 papers for the selection process. Following the identification stage, a screening was conducted to eliminate duplicate documents. Certainly, a journal or conference-published work may be archived in at least one research database. In this stage, 708 papers were retained after screening. The last step was to apply the eligibility criteria to select the best articles according to the Sections 2.1.3 and 2.1.4 Table 2. In the first screening, 97 articles were kept and 611 were rejected based on their titles; in the subsequent screening, 48 papers were retained and 49 were rejected based on their abstracts. After a thorough reading of each manuscript, 33 were ultimately chosen for study, while 15 were discarded.

**Table 2.** Exclusion criterion description.

| Criteria | Description |
| --- | --- |
| Date | The publications included for this research were screened between 1 January 2016 and 10 July 2022. The quantity of relevant articles to filter dictated the selection of this range. Given the fast advancement of deep learning models and machine learning. |
| Type of publications | filtering was performed on two categories of publications, papers published at international conferences and articles published in international journals. |
| Ranking | To determine the finest articles, we used the ranking count of papers published in journals systematically. This criteria was not applied to papers presented at conferences. We examined the rankings Q1, Q2, and Q3 for the publications in the journals. |
| Type of problem | Only articles on biomedical text or image-text classification were evaluated for this criteria. |
| Citations | This criteria was given less weight, particularly for articles published recently, such as those from 2021 and 2022 |

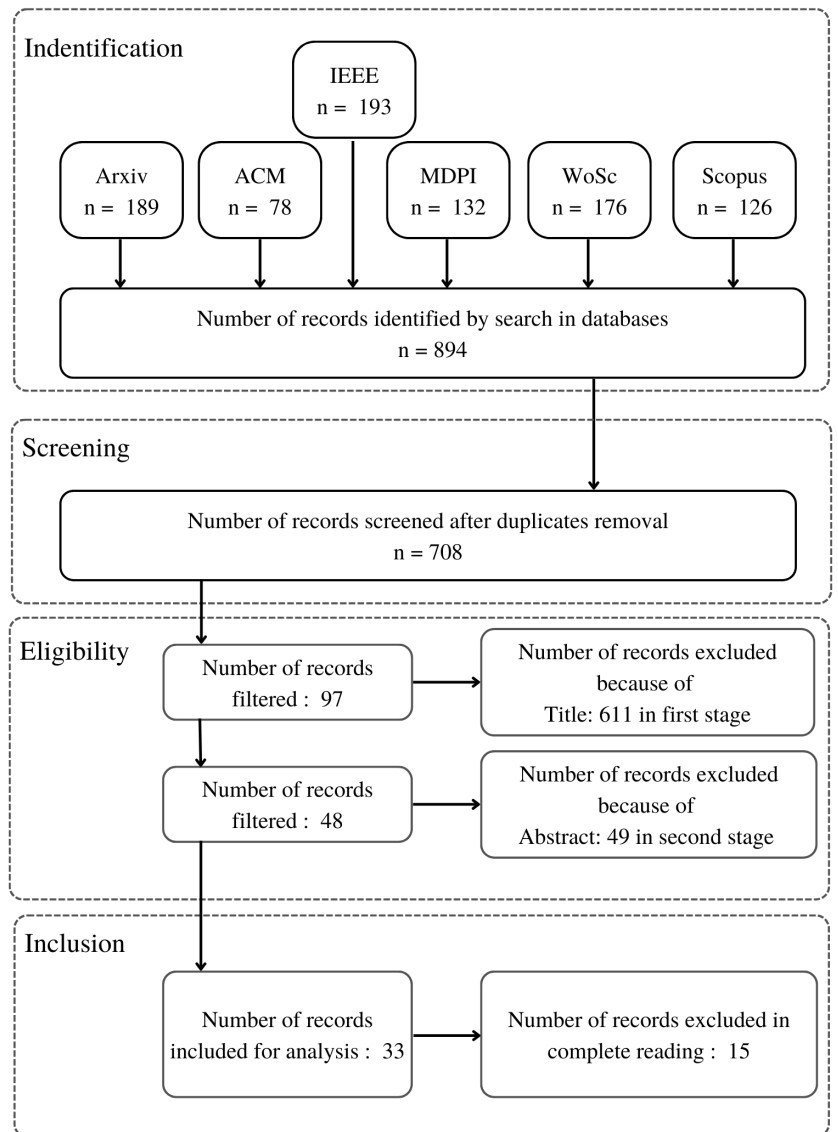

**Figure 1.** Paper selection flow diagram for text classification in biomedical domain.

## 2.2. Quality Metrics

The quality metrics in Table 3 were used to evaluate the relevance of each paper selected during the analysis see Table A2. The total score of 15 for the papers published in the international journal and the total score of 11 for those published in an international conference because the metric M11 for the ranking does not concern the conference papers. The following ratings have been defined for both types of papers: For journal papers, the paper is rating Excellent if the score is between 13–15; between 10–12 very good; between 7–9 good; between 3–6 sufficient and between 0–2 deficient. For the conference papers, the paper is rating Excellent if the score is between 9–11; between 6–8 very good; between 4–5 good, if this 3, sufficient and between 0–2 deficient.

**Table 3.** Quality metrics of paper selection.

| Category Metric | Metric | Description | Value | Weight |
|---|---|---|---|---|
| Metrics based on the text content of the paper (5 points) | M1 | Provide a clear and balanced summary according to the context of the problem solved in the paper | (No/Yes) [0,1] | 1 |
| | M2 | Provide details of the model's performance metrics and the entire evaluation process | [0,1] | 1 |
| | M3 | Implement one or more medical text classification models | [0,1] | 1 |
| | M4 | compares the results with other similar work and presents the limitations of this work | [0,1] | 1 |
| | M5 | Contains a deep and rich discussion on the result obtained | [0,1] | 1 |
| Other quality metrics (10 points ) | M6 | Innovation | [0,1] | 1 |
| | M7 | The dataset used in the research is a benchmark or it has been made publicly available | [No,Yes] (0–1) | 1 |
| | M8 | Performance (Accuracy) | Regarding the performance, if the percentage of quality of result is between 60–70% (0.5), between 71–80% (1), between 81–90% (1.5) and 91% + (2) otherwise (0) | 2 |
| | M9 | Citation | If the paper is cited 0 times (0), 1–4 times (0.5) and cited 6+ (1) | 1 |
| | M10 | Availability of source code | [0,1] | 1 |
| | M11 | Journal ranking | If rank = Q1 then (4), rank = Q2 then (3) rank = Q3 then (2) and if rank = Q4 then 1 | 4 |

## 3. Results

All the papers selected following the steps of the flow diagram in Figure 1 were included in the analysis. Table A1 summarizes the selections made in this review paper. All the metrics of Table 3 were applied to evaluate the selected papers, and the result of this evaluation is in Table 4. The whole evaluation process is presented in Table A2 in the Appendix A. In addition, to answer the questions in Table 1, the evaluation of the different text classification databases used in each selected paper was conducted in order to discover new challenges in the data and their influence in building the models. Finally, an evaluation of the frequency distribution of the selected papers by location, publication database, and type (Journal/Conference) was done, followed by an evaluation of the frequency distribution by ranking and year of publication.

### 3.1. Quality Metric Result

To make sure that the evaluations of each article's quality parameters were correct, the right measurements were taken using the defined indicators Table 3. Each article was ranked on a scale from deficient to excellent based on how much it added to our systematic review. They were judged based on the degree of innovation, the details of the proposal, validation, results and analysis, ranking, and the number of citations. Table A2 shows how each article did in terms of the score, and Table 4 gives a full summary of Table A2.

**Table 4.** Metric result.

| Score | No Journal | No Conference | Total |
|---|---|---|---|
| Excellence | 3 | 6 | 9 |
| Very good | 14 | 7 | 21 |
| Good | 2 | 1 | 3 |
| Sufficient | 0 | 0 | 0 |
| Deficient | 0 | 0 | 0 |
| Total | 19 | 14 | 33 |

*3.2. Text Classification Methods Performance According Datasets Used*

The best approaches in relation to the database were identified in two ways, based on one of the performance indicators such as accuracy, precision, recall, and F1-score. First, papers that utilized the same datasets were grouped together, and then all publications were grouped together. Two datasets that used more than one publication were identified such as: **MIMIC III** with five papers and **AIM** had two papers. With MIMIC III, the BioBERT approach in [7] has an Accuracy of 90.05% and is regarded the best method for this classification, whereas the LSTM method in [8] gets a score of 91.00%. In the publication [9], the BIGRU technique achieves 97.73% of accuracy on the AIM dataset. The synthesis is shown in Table 5.

**Table 5.** Performance of the most frequent text classification methods and database used.

| Methods | Dataset | Accuracy | Precision | Recall | F1-Score |
|---|---|---|---|---|---|
| TAGS [10] | MIMIC-III dataset | 82.00% | - | - | - |
| SWAM-text CNN [11] | MIMIC-III dataset | - | - | - | 60.00% |
| BioBERT [7] | MIMIC-III database | 90.05% | 77.37% | | 48.63% |
| BERT-base [12] | MIMIC III dataset | 82.30% | - | - | 82.20% |
| LSTM [8] | MIMIC-III dataset | - | - | - | 91.00% |
| QC-LSTM; BiGRU [13] | AIM dataset | 96.72% | - | - | - |
| BIGRU [9] | AIM dataset | **97.73%** | - | - | - |

Considering only the performance measurement values of the different classification techniques in general [14], without basing them on the direct comparison with the data used and their statistical distribution, the problem to be solved, we observe that, BERT-based technique in [15–17], GRU [9,13], BiGRU [9,13] and LSTM [8,18] produced a good performance on most of the problems studied in the different papers identified in our study. In addition, the methods that present the good performance but represented only once in the papers studied, are Random forest [19], CNN-MHA-BLSTM [20], Double-channel (DC-LSTM) [21], MT-MI-BiLSTM-ATT [22] and QC-LSTM [13] Table 6.

**Table 6.** Performance obtained on different text classification methods used in each paper.

| Methods | Dataset | Accuracy | Precision | Recall | F1-Score |
|---|---|---|---|---|---|
| TAGS [10] | MIMIC-III | 82.00% | - | - | - |
| SWAM-text CNN [11] | MIMIC-III full dataset; MIMIC-III 50 dataset | - | - | 60.00% | |
| BioBERT [7] | MIMIC-III database | 90.05% | 77.37% | | 48.63% |
| BERT-base [12] | PubMed abstract; MIMIC III | 82.30% | - | - | 82.20% |
| LSTM [8] | MIMIC-III; CSU dataset | - | - | - | 91.00% |
| QC-LSTM; BiGRU [13] | Hallmarks dataset; AIM dataset | 96.72% | - | - | - |
| BIGRU [9] | TCM—Traditional Chinese medicine dataset; CCKS dataset; Hall-marks—corpus dataset; AIM—Activating invasion and metastasis dataset | 97.73% | - | - | - |
| Conv-LSTM [23] | EMR text data (benchmark) | 83.30% | - | - | - |
| MT-MI-BiLSTM-ATT [22] | EMR data set comes from a hospital (benchmark) | 93.00% | - | - | 87.00% |
| SVM (Sigmoid Kernel) [24] | EMR data from outpatient visits during 2017 to 2018 at a public hospital in Surabaya City, Indonesia (benchmark) | 88.40% | 81.28% | 76.46% | 78.80% |
| BERT [15] | THUCNews; iFLYTEK | 96.63% | 96.64% | 96.63% | 96.61% |
| BERT-based [16] | COVID-19 fake news dataset" by Sumit Bank; extremist-non-extremist dataset | 99.71% | 98.82% | 97.84% | 98.33% |
| LSTM [18] | SQuAD | | 98.00% | 98.00% | 98.00% |
| MedTS [25] | MIMICSQL | 88.00% | - | - | - |
| CNN [26] | DingXiangyisheng's question and answer module (benchmark) | 86.28% | - | - | - |
| CRNN [27] | iDASH dataset; MGH dataset | - | - | - | 84.50% |
| Double-channel (DC-LSTM) [21] | cMedQA medical diagnosis dataset; Sentiment140 Twitter dataset | 97.20% | 91,80% | 91.80% | 91.00% |
| CNN Based model [28] | EMR Progress Notes from a medical center (benchmark) | 58.00% | 58.20% | 57.90% | 58.00% |
| BidirLSTM [29] | clinical nursing shift notes (benchmark) | - | - | - | - |
| Random forest [19] | Text dataset from NHLS-CDW | 95.25% | 94.60% | 95.69% | 95.34% |
| SVM [30] | Medical records from from digital health (benchmark) | 80.00% | - | - | - |
| CNN-MHA-BLSTM [20] | EMR texte dataset (benchmark) | 91.99% | - | - | 92.03% |
| MLP [31] | EMR dataset (benchmark) | 82.00% | - | - | 82.00% |
| MobileNetV2 [32] | RVL-CDIP dataset | - | - | - | 82.00% |
| Med2Vec [33] | CHOA dataset | - | - | 91.00% | - |
| biGRU [34] | RCV1/RCV2 dataset | - | - | - | 84.00% |
| Capsule+LSTM [35] | Chinese electronic medical record dataset | - | - | - | 73.51% |
| BioLinkBERT [36] | MedQA-USMLE; MMLU-professional medicine | 50.00% | - | - | - |
| Bert-based [17] | Harvard obesity 2008 challenge dataset | 94.70% | - | - | - |

*3.3. Frequency Result According Geographical Distribution and Type of Publication*

As we can see in Table 7, several studies based on text classification were carried out in Asia with a percentage of 51.5, which is half of all the papers analyzed in our research. With 6.1 percent, Africa has a low representation papers, whereas America and Europe both have 21.2 percent. It is also shown in this study that it has 57.6% of papers published in journals and 42.4% published in conferences. In Figure 2, we present the different frequencies of the selected papers according to regions, continents, search database and type of publications. The most frequented region with published studies on medical text classification was the Eastern Asia region. In addition, among the search databases Web Of Science was the most frequented database after filtering.

**Table 7.** Number and frequency of research database, conference or journal and by geographical distribution of publication.

| Parameters | Category | Frequency | |
|---|---|---|---|
| | | **No. Papers** | **Percentage** |
| **Location** | Southern Africa | 1 | 3.0% |
| | **Africa** | **1** | **3.0%** |
| | Eastern Asia | 13 | 39.4% |
| | Southern Asia | 2 | 6.1% |
| | Western Asia | 1 | 3.0% |
| | South-Eastern Asia | 1 | 3.0% |
| | **Asia** | **17** | **51.5%** |
| | Northern Europe | 3 | 9.1% |
| | Eastern Europe | 1 | 3.0% |
| | Southern Europe | 2 | 6.1% |
| | Western Europe | 2 | 6.1% |
| | **Europe** | **8** | **24.3%** |
| | Northern America | 7 | 21.2% |
| | **America** | **7** | **21.2%** |
| **Database** | Arxiv | 7 | 21.2% |
| | ACM | 2 | 6.1% |
| | MDPI | 2 | 6.1% |
| | WoSc | 10 | 30.3% |
| | Scopus | 4 | 12.1% |
| | IEEE | 8 | 24.2% |
| **Type of publication** | Conference | 14 | 42.4% |
| | Journal | 19 | 57.6% |

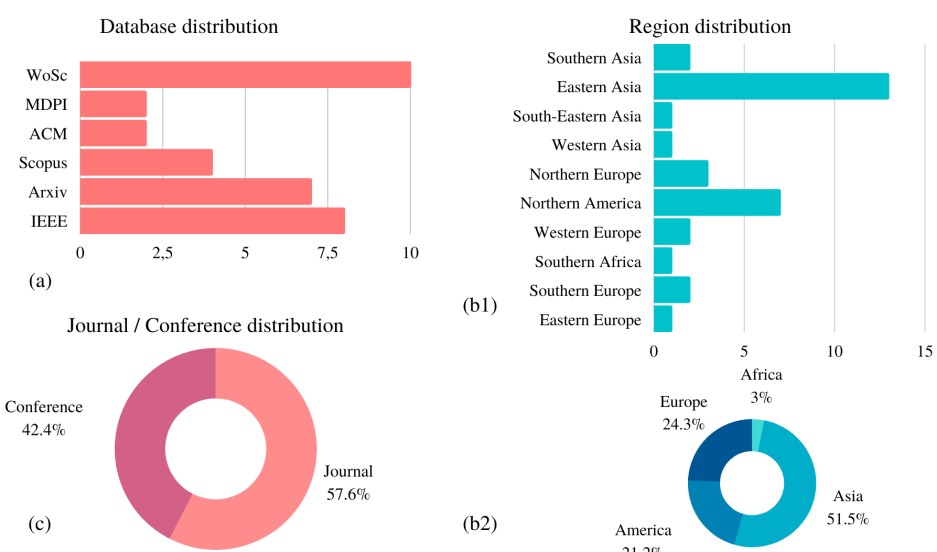

**Figure 2.** Frequency of research database, conference or journal and by geographical distribution of publication. (**a**) is the distribution of the different databases from which we collected papers in this study. (**b1,b2**) represent respectively the distribution of the selected papers by region and by continent. (**c**) the distribution of papers by conference and journal.

### 3.4. Paper Publication Map by Country

The map in Figure 3 describes the degree of contribution of countries in Artificial Intelligence (NLP) in biomedical text classification from 1 January 2016 to 10 July 2022. China largely dominates in this study, followed by the USA, this result coincides with the result published in Artificial Intelligence Index Report 2022 [37].

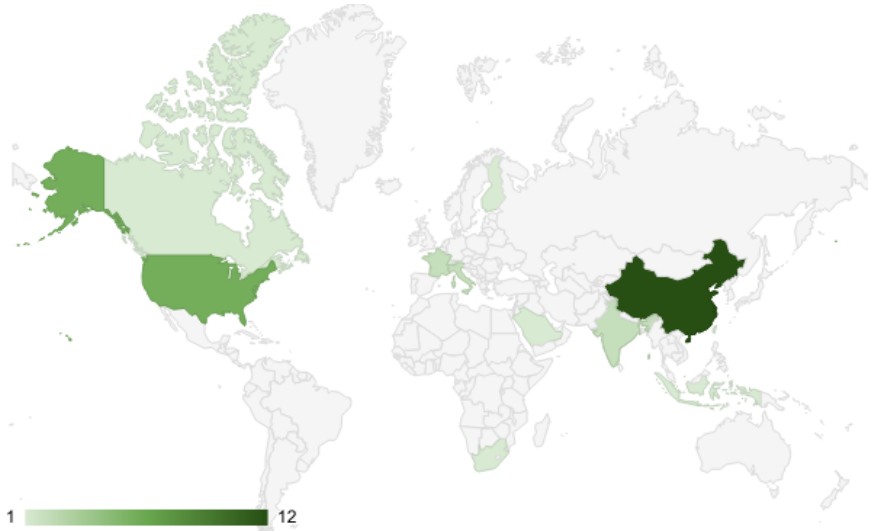

**Figure 3.** Degree of contribution of countries in Artificial Intelligence (NLP) in biomedical text classification.

### 3.5. Frequency Result According Year and Journal Ranking

Table 8 shows the frequency and number of papers per year and per ranking. As mentioned above, the time range considered for the selection of articles for analysis in this systematic review was from 1 January 2016 to 10 July 2022. The year 2020 counted 11 papers and represents 33.3% of the papers compared to other years with low representatives in the classification of biomedical texts. In addition, the ranking was considered as one of the major eligibility criteria of papers for analysis in the case of papers published in journals. All the papers whose category is none are published in an international conference.

Considering the ranking, more of the selected papers, i.e., 12 out of 19 papers published in journals, were of Q1. Figure 4 presents the different frequencies in the analysis for the year and the ranking distribution.

**Table 8.** Number and frequency of year and paper ranking.

| Parameters | Category | Frequency | |
| --- | --- | --- | --- |
| | | No. Papers | Percentage |
| **Year** | 2016 | 1 | 3.0% |
| | 2017 | 3 | 9.1% |
| | 2018 | 1 | 3.0% |
| | 2019 | 5 | 15.1% |
| | 2020 | 11 | 33.3% |
| | 2021 | 8 | 24.3% |
| | 2022 | 4 | 12.1% |
| **Paper ranking** | Q1 | 12 | 36,4% |
| | Q2 | 3 | 9.1% |
| | Q3 | 3 | 9.1% |
| | Conference | 15 | 45.5% |

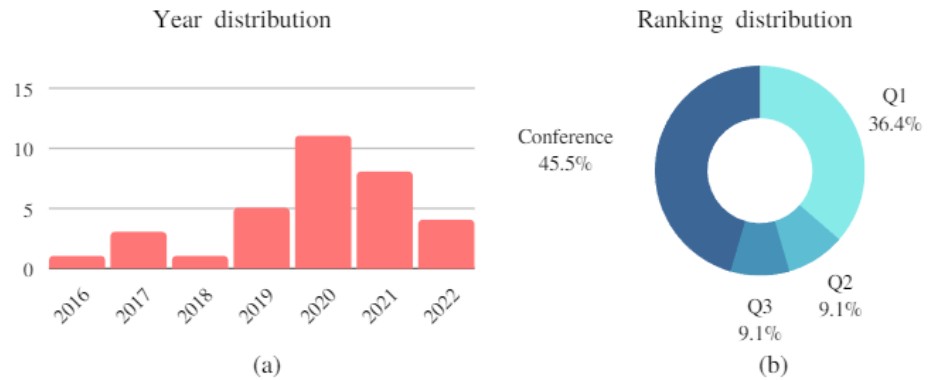

**Figure 4.** (**a**) Represent the frequency by year and (**b**) the distribution by conference and paper ranking.

## 4. Discussion

Text classification in biomedical field plays an important role in the rapid search (diagnosis) of a disease from the patient record, hospital administration, and even the treatment appropriate for a specific case, as the volume of patient medical records continues to increase significantly. Each year, new classification methods with high classification accuracy are proposed, while the performance of older [38–40] NLP methods is enhanced through the use of alternative approaches such as optimization and other type of algorithm based on transformers architecture [12,40–42] and XLNet [43], data-centric technique and many others. The data-centric technique presents a challenge in enhancing the performance of biomedical text classification methods [44]. The observation is that the majority of methods have been pre-trained with text databases in a generic context without any prior specificity. In other words, a model that has been pre-trained with biomedical data will adapt better when re-trained with new data for a biomedical domain. In this context, we discuss the data-centric problem, which must be a key consideration when developing models tailored to specific case. Another challenge in the classification of biomedical texts is the data quality. We found two kinds of datasets in the articles we looked at: those made public by research institutes and labs [7,9,13,15–17,21], and those that any reference

(benchmark) could use without more information. When training the models to give good results, it is important to think about how good the data [45] are. This quality can be made sure of by thinking about the whole process of collecting and preprocessing the data until it is ready to be used for classification tasks.

Before performing the classification task, biomedical texts can be derived from a variety of sources [46,47]. We find data in medical reports that are already in text form, as well as notes taken by doctors or nurses during consultations that are scanned images. Depending on the context of the problem and the goal to be achieved, several approaches can be used with these types of data. Alternatively, the data can be represented in both formats, or a radio image is accompanied by text that explains the image. Depending on the expected result, several methods can be combined in the text classification process in image-text data [13]. To complete these tasks, methods based on CNN architectures [48,49] are frequently used [13,50].

The classification of biomedical texts is involved in several important problems that physicians are expected to solve. These issues can sometimes be large challenges in multiple steps. This can be conducted in diagnosis [11,28], patient treatment [11], or even effective representations of concepts such as diagnoses, drugs, procedure codes, and patient visits [33], as well as in the quick search of a document or disease classification [23]. Pathologies from clinical notes [23] and much more In all of these ways, it is harder to classify texts in the biomedical field than in other fields in general. This is because biomedical texts include both medical records and medical literature, which are both important sources of clinical information. However, medical texts have hard-to-understand medical terms and measurements that cause problems with high dimensionality and a lack of data [9]. All of these problems are very important when it comes to the task of classifying biomedical text.

In the biomedical text classification task, as in most classification problems in general [51], the model evaluation metrics are the same. In all the papers studied in our systematic review, the metrics identified are Accuracy, Recall, F1-score, Precision, Average precision, Average recall, and Average F1-score. These metrics are the most commonly used to evaluate text classification models. As in this study, the different methods used in each paper analyzed, used at least one of these metrics except for one paper [52] which used Spearman.C.C. metric [53].

## 5. Conclusions and Perspectives

This study discusses the various challenges in the task of biomedical text classification by focusing on several aspects such as the challenge in method performance, discovering the structure of biomedical data for the classification task, listing the various problems and challenges that text classification can solve in the biomedical domain, and finally reviewing the most commonly used metrics to evaluate biomedical text classification methods. We discovered two significant issues linked to the approaches utilized for biomedical text classification by reviewing the various literature chosen for examination in this research. First, there is the data-centric, which is explained by the fact that most transfer learning using pre-trained techniques employ a dataset of broad text classification settings. However, the biomedical issue includes various medical words that may be classified as process, therapy, medicine, and diagnosis. Because the contextual representation of medical language is quite poor in the general context, this already creates a contextual challenge when training to generate the best outcomes. This necessitates only training models on a huge number of biological data in order to execute transfer learning more correctly. There are certain approaches that are exclusively trained with biomedical databases, such as BioBERT [7] and BioLinkBERT [17], but the task remains to study as many ways as possible with just biomedical databases to enhance biomedical text classification outcomes. This is the first problem that affects how well the text classification methods work in biomedical domain.

Another issue to consider is data quality. We found two types of datasets in the articles we looked at: those made public by research institutes and labs, and those accessed by any reference (benchmark) without more information. The quality of the data is a key factor to

consider while training the models to deliver good outcomes. This quality may be assured by considering the whole collecting and pre-processing process until the data set is ready as an usable source for classification tasks. Several other challenges can be described by taking into account several aspects that we have not addressed in this work. Some of the challenges we have discussed are the most common ones in our overall study.

In the perspective, to significantly advance research in the biomedical field, it is preferable to make well-preserved and verified data more widely available in order to assist research and overcome data quality [54–56] in biomedical classification challenges. Because of domain drifts among different institutes, the cooperation between research laboratories, universities and other research entities, would be an action to be strengthened in order to create a great network of scientific sharing of scarce resources such as data to advance research. Joint work sessions between domain experts should be a good procedure to validate the dataset as a common resource for scientific research of text classification. Finally, a policy of simplification of data sharing, which is often confidential, would be an essential point among many others to be defined to answer the problem of data deficiency. Most of the models used in the papers selected in this study are based on Deep learning. The interpretability of robust models is an important aspect in clinical research. Accuracy and reliability are also important aspects in biomedical research field. Whether one uses simple models based on statistical learning or robust models based on Deep learning, whatever their performance, the interpretability and reliability aspect would be very important to take into account, to validate the results for a clinical research.

**Author Contributions:** Conceptualization, C.Y.K., A.C.-V and B.G.-Z.; methodology, C.Y.K. and A.C.-V.; formal analysis, C.Y.K.; investigation, C.Y.K.; writing—original draft preparation, C.Y.K.; writing—review and editing, C.Y.K., A.C.-V. and B.G.-Z.; supervision, B.G.-Z.; All authors have read and agreed to the published version of the manuscript.

**Funding:** This research received no external funding.

**Institutional Review Board Statement:** Not applicable.

**Informed Consent Statement:** Not applicable.

**Data Availability Statement:** Not applicable.

**Acknowledgments:** The authors thank and acknowledge the eVida research group of the University of Deusto, recognized by the Basque Government with the code IT1536-22, and the ICCRF Colon Cancer Challenge for its untiring support and commitment to providing us with the resources necessary to carry out the study, until its finalization.

**Conflicts of Interest:** The authors declare no conflict of interest.

# Appendix A

**Table A1.** Summary of papers filtering aspects.

| P. | Year | J/C | Loc | Database | Methods | Dataset | Best Method | Metric (Best) | Rank | Cite |
|---|---|---|---|---|---|---|---|---|---|---|
| [10] | 2020 | J | India | Scopus | Fuzzy similarity-based data cleansing approach, supervised multi-label classification models, MLP, KNN, KNN as OvR, AGgregated using fuzzy Similarity (TAGS) | MIMIC-III | TAGS | Ac: 82.0 | Q1 | 18 |
| [23] | 2020 | J | India | IEEE | MLP, ConvNet, LSTM, Bi-LSTM, Conv-LSTM, Seg-GRU | EMR text data (benchmark) | Conv-LSTM | Ac: 83.3 | Q1 | 7 |
| [22] | 2020 | J | China | IEEE | BiLSTM, CNN, CRF layer. In particular, they used BiLSTM and CNN to learn text features and CRF as the last layer of the model; MT-MI-BiLSTM-ATT | EMR data set comes from a hospital (benchmark) | MT-MI-BiLSTM-ATT | Ac: 93.0 F1: 87.0 | Q1 | 3 |
| [57] | 2021 | C | China | IEEE | ResNet; BERT-BiGRU; ResNet-BERTBiGRU | Text-image data (benchmark) | ResNet-BERTBiGRU | Mavg.P: 98.0 Mavg.R: 98.0 Mavg.F1: 98.0 | None | 0 |
| [58] | 2021 | C | Indonesia | IEEE | SVM (Linear Kernel); SVM (Polynomial Kernel); SVM (RBF Kernel); SVM (Sigmoid Kernel) | EMR data from outpatient visits during 2017 to 2018 at a public hospital in Surabaya City, Indonesia (benchmark) | SVM (Sigmoid Kernel) | R: 76.46; P: 81.28; F1: 78.80; Ac: 91.0 | None | 0 |
| [24] | 2020 | C | China | IEEE | GM; Seq2Seq; CNN; LP; HBLA-A (This model can be seen as a combination of BERT and BiLSTM.) | ARXIV Academic Paper Dataset (AAPD); Reuters Corpus Volume I (RCV1-V2) | BLA-A | Micro-P: 90.6; Micro-R: 89.2; Micro-F1: 89.9 | None | 1 |
| [15] | 2021 | C | China | IEEE | Text CNN; BERT; ALBERT | THUCNews; iFLYTEK | BERT | Ac: 96.63; P: 96.64; R: 96.63; F1: 96.61 | None | 0 |
| [16] | 2022 | J | Saudi Arabia | Scopus | BERT-base; BERT-large; RoBERTa-base; RoBERTa-large; DistilBERT; ALBERT-base-v2; XLM-RoBERTa-base; Electra-small; and BART-large | COVID-19 fake news dataset" by Sumit Bank; extremist-non-extremist dataset | BERT-base | Ac: 99.71; P: 98.82; R: 97.84; F1: 98.33 | Q3 | 3 |
| [18] | 2020 | C | UK | WoSc | LSTM; Multilingual; BERT-base; SCIBERT; SCIBERT 2.0 | SQuAD | LSTM | P: 98.0; R: 98.0; F1: 98.0 | None | 10 |

**Table A1.** *Cont.*

| P. | Year | J/C | Loc | Database | Methods | Dataset | Best Method | Metric(best) | Rank | Cite |
|---|---|---|---|---|---|---|---|---|---|---|
| [13] | 2021 | J | China | WoSc | CNN, LSTM, BiLSTM, CNN-LSTM, CNN-BiLSTM, logistic regression, naïve Bayesian classifier (NBC), SVM, and BiGRU. QC-LSTM; BiGRU | Hallmarks dataset; AIM dataset | QC-LSTM | AC: 96.72 | Q3 | 1 |
| [25] | 2021 | J | China | WoSc | Seq2Seq; SQLNet; PtrGen; Coarse2Fine; TREQS; MedTS | MIMICSQL | MedTS | AC: 88.0 | Q2 | 0 |
| [26] | 2029 | J | China | WoSc | CNN; LSTM | DingXiangyisheng's question and answer module (benchmark) | CNN | AC: 86.28 | Q1 | 1 |
| [27] | 2027 | J | USA | WoSc | Tf-Idf CRNN | iDASH dataset; MGH dataset | CRNN | AUC: 99.1; F1: 84.5 | Q1 | 59 |
| [21] | 2027 | J | China | WoSc | CNN; LSTM; CNN-LSTM; GRU; DC-LSTM | cMedQA medical diagnosis dataset; Sentiment140 Twitter dataset | DC-LSTM | Ac: 97.2; P: 91.8; R: 91.8; F1: 91.0 | Q3 | 1 |
| [28] | 2020 | J | Taiwan | WoSc | CNN; CNN Based model | EMR Progress Notes from a medical center (benchmark) | CNN Based model | Ac: 58.0; P: 58.2; R: 57.9; F1: 58.0 | Q1 | 2 |
| [9] | 2019 | J | China | Scopus | CNN; RCNN; LSTM; AC-BiLSTM; SVM; Logistic-Regression | TCM—Traditional Chinese medicine dataset; CCKS dataset; Hallmarks—corpus dataset; AIM—Activating invasion and metastasis dataset | BIGRU | Hallmarks, Ac: 75.72; TCM, Ac: 89.09; CCKS, Ac: 93.75; AIM, Ac: 97.73 | Q2 | 15 |
| [59] | 2021 | J | China | WoSc | RoBERTa; ALBERT; transformers-sklearn based | TrialClassifcation, BC5CDR, DiabetesNER, and BIOSSES | transformers-sklearn based | Mavg-F1: 89.03 | Q1 | 2 |
| [7] | 2020 | C | UK | WoSc | BioBERT; Bert | MIMIC-III database | BioBERT | Ac: 90.05; Precision: 77.37; F1: 48.63 | None | 0 |
| [29] | 2019 | C | Finland | IEEE | BidirLSTM, LSTM, CNN, fastText, BoWLinearSVC, RandomForest, Word Heading Embeddings, Most Common, Random | clinical nursing shift notes (benchmark) | BidirLSTM | Avg-R: 54.35 | None | 3 |

**Table A1.** *Cont.*

| P. | Year | J/C | Loc | Database | Methods | Dataset | Best Method | Metric(best) | Rank | Cite |
|---|---|---|---|---|---|---|---|---|---|---|
| [19] | 2021 | J | South Africa | MDPI | Random forest, SVMLinear, SVMRadial | text dataset from NHLS-CDW | Random forest | F1: 95.34; R: 95.69 P: 94.60 Ac: 95.25 | Q2 | 2 |
| [30] | 2020 | J | Italy | Scopus | SVM | Medical records from from digital health (benchmark) | SVM | Mavg-P: 88.6; Mavg-Ac: 80.0 | Q1 | 27 |
| [8] | 2020 | J | UK | WoSc | LSTM; LSTM-RNNs; SVM, Decision Tree; RF | MIMIC-III; CSU dataset | LSTM | F1: 91.0 | Q1 | 13 |
| [20] | 2020 | C | USA | ACM | CNN-MHA-BLSTM; CNN, LSTM | EMR texte dataset (benchmark) | CNN-MHA-BLSTM | Ac: 91.99; F1: 92.03 | None | 22 |
| [31] | 2019 | C | USA | IEEE | MLP | EMR dataset (benchmark) | MLP | Ac: 82.0; F1: 82.0 | None | 1 |
| [12] | 2019 | C | USA | Arxiv | BERT-base, ELMo, BioBERT | PubMed abstract; MIMIC III | BERT-base | Ac: 82.3 | None | 0 |
| [32] | 2020 | J | France | Arxiv | MLP, CNN CNN 1D, MobileNetV2, MobileNetV2 (w/ DA) | RVL-CDIP dataset | MobileNetV2 | F1: 82:0 | Q1 | 55 |
| [33] | 2016 | C | USA | ACM | Med2Vec | CHOA dataset | Med2Vec | R: 91.0 | None | 378 |
| [52] | 2018 | C | Canada | Arxiv | word2vec, Hill, dict2vec | MENd dataset; SV-d dataset | word2vec | Spearman.C.C: 65.3 | None | 37 |
| [34] | 2017 | C | Switzerland | Arxiv | biGRU, GRU, DENSE | RCV1/RCV2 dataset | biGRU | F1: 84.0 | None | 34 |
| [11] | 2021 | J | China | Arxiv | Logistic regression; SWAM-CAML; SWAM-text CNN | MIMIC-III full dataset; MIMIC-III 50 dataset | SWAM-text CNN | F1: 60.0 | Q1 | 6 |
| [35] | 2022 | J | China | MDPI | LSTM, CNN, GRU, Capsule+GRU, Capsule+LSTM | Chinese electronic medical record dataset | Capsule+LSTM | F1: 73.51 | Q2 | 2 |
| [36] | 2022 | C | USA | Arxiv | BERTtiny; LinkBERTtiny, GPT-3, BioLinkBERT, UnifiedQA | MedQA-USMLE; MMLU-professional medicine | BioLinkBERT | Ac: 50.0 | None | 4 |
| [17] | 2022 | J | USA | Arxiv | CNN, LSTM, RNN, GRU, Bi-LSTM, Transformers, Bert-based | Harvard obesity 2008 challenge dataset | Bert-based | Ac: 94.7 | Q1 | 0 |

**Table A2.** Results of the application of the eligibility criteria to the filtered papers.

| P. | M1 | M2 | M3 | M4 | M5 | M6 | M7 | M8 | M9 | M10 | M11 | Result |
|---|---|---|---|---|---|---|---|---|---|---|---|---|
| [10] | 1 | 1 | 1 | 0 | 1 | 1 | 1 | 1 | 1 | 0 | 4 | 12/15 |
| [23] | 1 | 1 | 1 | 0 | 1 | 1 | 1 | 1 | 1 | 0 | 4 | 12/15 |
| [22] | 1 | 1 | 1 | 0 | 1 | 1 | 0 | 2 | 1 | 0 | 4 | 12/15 |
| [57] | 1 | 1 | 1 | 0 | 1 | 0 | 0 | 2 | 0 | 0 | Conf. | 6/11 |
| [58] | 1 | 1 | 0 | 0 | 1 | 0 | 0 | 2 | 0 | 0 | Conf. | 5/11 |
| [24] | 1 | 1 | 1 | 1 | 1 | 0 | 1 | 2 | 1 | 0 | Conf. | 9/11 |
| [15] | 1 | 1 | 1 | 0 | 1 | 0 | 0 | 2 | 0 | 0 | Conf. | 6/11 |
| [16] | 1 | 1 | 1 | 0 | 1 | 0 | 0 | 2 | 0.5 | 0 | 2 | 8.5/15 |
| [18] | 1 | 1 | 1 | 1 | 1 | 0 | 1 | 2 | 1 | 0 | Conf. | 9/11 |
| [13] | 1 | 1 | 1 | 1 | 1 | 0 | 1 | 2 | 0.5 | 1 | 2 | 11.5/15 |
| [25] | 1 | 1 | 1 | 0 | 1 | 0 | 1 | 2 | 1 | 0 | 3 | 11/15 |
| [26] | 1 | 1 | 1 | 1 | 1 | 0 | 1 | 1.5 | 0.5 | 0 | 4 | 12/15 |
| [27] | 1 | 1 | 1 | 1 | 1 | 0 | 1 | 2 | 1 | 1 | 4 | 14/15 |
| [21] | 1 | 1 | 1 | 1 | 1 | 0 | 1 | 2 | 0.5 | 0 | 3 | 11/15 |
| [28] | 1 | 1 | 0 | 1 | 1 | 0 | 1 | 0.5 | 0.5 | 0 | 4 | 9/15 |
| [9] | 1 | 1 | 1 | 1 | 1 | 0 | 1 | 2 | 1 | 0 | 3 | 12/15 |
| [59] | 1 | 1 | 1 | 1 | 1 | 0 | 1 | 1.5 | 0.5 | 0 | 4 | 12/15 |
| [7] | 1 | 1 | 1 | 1 | 0 | 0 | 1 | 1.5 | 0 | 0 | Conf. | 6.5/11 |
| [29] | 1 | 1 | 1 | 1 | 1 | 0 | 1 | 0 | 0.5 | 0 | Conf. | 6.5/11 |
| [19] | 1 | 1 | 1 | 1 | 1 | 0 | 1 | 2 | 0.5 | 0 | 3 | 11.5/15 |
| [30] | 1 | 1 | 0 | 1 | 1 | 1 | 0 | 1.5 | 1 | 0 | 4 | 11.5/15 |
| [8] | 1 | 1 | 1 | 1 | 1 | 0 | 0 | 2 | 1 | 0 | 4 | 12/15 |
| [20] | 1 | 1 | 1 | 1 | 1 | 1 | 0 | 2 | 1 | 0 | Conf. | 9/11 |
| [31] | 1 | 1 | 0 | 1 | 1 | 1 | 0 | 1.5 | 0.5 | 0 | Conf. | 7/11 |
| [12] | 1 | 1 | 1 | 1 | 1 | 0 | 0 | 0 | 0 | 1 | Conf. | 6/11 |
| [32] | 1 | 1 | 1 | 1 | 1 | 0 | 2 | 1.5 | 1 | 1 | 4 | 14.5/15 |
| [33] | 1 | 1 | 0 | 1 | 1 | 1 | 2 | 1 | 1 | 1 | Conf. | 10/11 |
| [52] | 1 | 1 | 1 | 1 | 1 | 1 | 0.5 | 1 | 1 | 1 | Conf. | 9.5/11 |
| [34] | 1 | 1 | 1 | 1 | 1 | 1 | 1.5 | 1 | 1 | 1 | Conf. | 10.5/11 |
| [11] | 1 | 1 | 1 | 1 | 1 | 0 | 0 | 1 | 1 | 1 | 4 | 12/15 |
| [35] | 1 | 1 | 1 | 1 | 1 | 0 | 1 | 1 | 0.5 | 1 | 3 | 11/15 |
| [36] | 1 | 1 | 1 | 1 | 1 | 1 | 1 | 0 | 0.5 | 1 | Conf. | 8/11 |
| [17] | 1 | 1 | 1 | 1 | 1 | 0 | 1 | 2 | 0 | 1 | 4 | 13/15 |

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
