# Peer review of "Natural Language Processing Techniques for Text Classification of Biomedical Documents: A Systematic Review"

_information, doi:10.3390/info13100499_

Round 1
Reviewer 1 Report
In this paper, the authors reviewed the NLP techniques for text classification of biomedical documents. The objective of review is to analyze the literature on various problems of classification of medical texts of patients based on criteria such as: the quality of the evaluation metrics used, the different methods of machine learning applied, the different data sets, and to highlight the best methods in this type of problem. The authors explored two major issues linked to the methodology and data used for biomedical text classification. First, there is the data-centric challenge, followed by the data quality challenge.
There are the following issues which need to be taken care by the authors so that the paper becomes acceptable.
1. In the abstract, the sentence- "In order to overcome the various challenge associated with the process of biomedical text classification" is incomplete. Also, "various challenge" should be replaced by "various challenges".
2. Explore the Discussion section, by considering evaluation parameters used by different techniques.
Reviewer 2 Report
Please find the comments in the attachment

Reviewer 3 Report
The authors carried out a thorough review that follows the PRISMA methodology. But out of four research questions shown in Table 1, they answered only the last one (I assume, that by evaluation metrics the authors mean performace metrics shown e.g. in Table 5). Instead, they present geographical and temporal information about the papers which in my opinion is not relevant for understanding the research trends in biomedical text classification.
Another questionable issue is the significant reduction of analysed papers from 894 to 33. So how general can be conclusions drawn from the rather small number of papers?
Round 2
Reviewer 3 Report
The authors considered my comments in the revised version.